# A Study on Optimization of Noise Reduction of Powered Vehicle Seat Movement Using Brushless Direct-Current Motor

**DOI:** 10.3390/s23052483

**Published:** 2023-02-23

**Authors:** Hyunju Lee, Dongshin Ko, Jaehyeon Nam

**Affiliations:** AI & Mechanical System Center, Institute for Advanced Engineering, Youngin-si 17180, Republic of Korea

**Keywords:** design of experiment, Monte Carlo simulation, robust design, optimization, noise reduction

## Abstract

In this paper, an optimal design model was developed to reduce noise and secure the torque performance of a brushless direct-current motor used in the seat of an autonomous vehicle. An acoustic model using finite elements was developed and verified through the noise test of the brushless direct-current motor. In order to reduce noise in the brushless direct-current motor and obtain a reliable optimization geometry of noiseless seat motion, parametric analysis was performed through the design of experiments and Monte Carlo statistical analysis. The slot depth, stator tooth width, slot opening, radial depth, and undercut angle of the brushless direct-current motor were selected as design parameters for design parameter analysis. Then, a non-linear prediction model was used to determine the optimal slot depth and stator tooth width to maintain the drive torque and minimize the sound pressure level at 23.26 dB or lower. The Monte Carlo statistical method was used to minimize the deviation of the sound pressure level caused by the production deviation of the design parameters. The result is that the SPL was 23.00–23.50 dB with a confidence level of approximately 99.76% when the level of production quality control was set at 3σ.

## 1. Introduction

Conventional direct-current (DC) motors with a brush are being replaced with high-efficiency, low-noise, high-ease-of-control, and high-durability brushless DC (BLDC) motors. Therefore, the development of BLDC motor drive system technology is critical to reduce energy consumption and produce eco-friendly vehicles. BLDC motors can be used to obtain low-power-consumption drive systems in automobiles [1,2,3]. Furthermore, automobile seats are being motorized to realize various motions because with the development of autonomous driving technology, indoor space is becoming critical. Domestic and foreign automobile and seat makers introduced concept cars equipped with various power-operated seats to emphasize the living space in the interior space of the vehicle [4]. High vehicle interior space can increase the degree of spatial freedom to realize various seat motions. Therefore, studies on maximizing passenger comfort [5] have investigated the use of BLDC as a driving source to realize different motions of a seat noiselessly.

In studies on BLDC motor noise, Kwon et al. [6] reported that increasing the number of slots reduces noise, and Jeong et al. [7] reported that among the torque ripple from the cogging torque and unbalanced excitation force, the radial force acting on the stator or rotor of a motor was the main cause of the noise. A torque ripple refers to the variability of torque generated during motor operation and is the major cause of vibration and noise. The main causes of torque ripple can be summarized as follows: the first cause is the cogging torque [8,9], which results from the magnetic force generated between the permanent magnet and the stator teeth when the rotator is rotated without applying a current. The second cause is the magnetic saturation [10] at the end of the teeth such that the magnetic flux density does not increase beyond a certain level even when the magnetic force continues to increase. The third cause is the counter electromotive force [11], which occurs when the sum of the phase current is inconsistent because the magnetic flux generated from the permanent magnet produces a voltage in the direction opposite to the stator winding as the rotator rotates. In addition to these electromagnetic effects, the structural effects of motors are major factors for vibration and noise. Lee et al. [12] confirmed that when the noise radiation frequency is proportional to the revolutions per minute (rpm) of the motor, the electromagnetic excitation force is relevant, but when it is not proportional to the rpm, the structural or acoustic mode is relevant.

Design of experiments (DoE) was used to optimize motor parameters. DoE is an ap-plication of statistics and can be used to obtain maximum information from the minimum number of experiments [13]. The method was developed by Ronald Fisher [14] for agricultural experiments and subsequently expanded to other industries such as the chemical industry after World War II [15]. Currently, this method is used in numerous research fields, such as medicine, engineering, psychology, and sociology [16,17,18,19]. Furthermore, the sensitivity and performance distribution of the design parameters were derived probabilistically using Monte Carlo simulation. Monte Carlo simulation is an algorithm that involves repeated random sampling to obtain mathematical results and is used in fields such as physics, medicine, finance, and artificial intelligence [20,21,22,23].

In this study, a design analysis process was proposed and verified to develop a seat motor for electric vehicles that improves motor drive torque and minimizes noise. Noise was measured through experiments, and the simulation model was verified based on frequency analysis. Partial factorial design was used to identify parameters with low influence, and response surface design was used for optimization. Noise optimization analysis was performed based on the acoustic mode. Additionally, Monte Carlo simulations were used to analyze the performance of the optimization for tolerance. Therefore, a process for noise optimization of BLDC motors was developed, and the proposed process is shown in Figure 1.

The paper is structured as follows: in Section 2, a BLDC motor design model and simulation model are described; in Section 3 numerical analysis for the motor noise is verified and design of experiments and the probabilistic method are conducted for the optimization of the design; in Section 4, there is a discussion of results, followed by the conclusion.

## 2. Simulation Model of BLDC

### 2.1. BLDC Motor Design Model

A BLDC motor consists of a permanent magnet rotor, winding coiled stator, and core case. Because the BLDC motor rotates the permanent magnet, changes in the complex flux can be controlled by varying the direction and timing of the current applied to the coiled stator to drive the motor. Thus, smooth rotation can be achieved by controlling the three-phase sinusoidal current. Furthermore, BDLC motors are typically used in automobile powered seats that require high output relative to the motor volume because they are highly reliable and efficient, and electrical and mechanical noises are low because of the absence of the brush on the rotor. Figure 2 displays the configuration of the BLDC motor used in this study, and Table 1 summarizes its specifications. 

### 2.2. Analysis Model of the BLDC Motor

#### 2.2.1. Magnetic Analysis Model of the BLDC Motor

The BLDC analysis model consisted of an electromagnetic analysis model and a noise analysis model, and the Flux and Simlab modules in the commercial analysis program HyperWorks were used for each analysis. Because no changes occurred in the shape of the motor in the longitudinal direction, the electromagnetic analysis model of the BLDC motor was simplified to a two-dimensional (2D) cross-sectional model and was constructed with 7418 elements and 15,466 nodes, as shown in Figure 3. Table 2 lists the property values of each component of the electromagnetic analysis model, and the three-phase current conditions for the electromagnetic analysis were applied according to the following equations, as illustrated in Figure 4.
(1)$$CON1=I_{m}\sin\left(\omega t+\gamma \right)$$
(2)$$CON2=I_{m}\sin\left(\omega t+\gamma -\frac{2\pi }{3}\right)$$
(3)$$CON3=I_{m}\sin\left(\omega t+\gamma -\frac{4\pi }{3}\right)$$
where $I_{m}$ is the maximum current of 4.64 A, *ω* is the frequency, *t* is the time, and *γ* is set at 45°.

The boundary condition for the BLDC motor electromagnetic analysis was set as an infinite box so that the magnetic field distribution of the analysis model was not affected by external factors. This condition can be expressed using the following equation:(4)$$\underset{r\to \infty }{\lim}\sqrt{r}\left[\frac{\partial E\left(r\right)}{\partial r}-ik_{e}E\left(r\right)\right]=0$$
where E is the electric field, r is the position vector, $k_{e}$($=\omega \sqrt{\mu_{0}\varepsilon_{0}}$) is the electromagnetic wave number, $\mu_{0}$ is the permeability of air, and $\varepsilon_{0}$ is the permittivity of air.

Under the assumption of a 2D magnetostatic field, the finite element analysis of the BLDC motor can be expressed in Maxwell’s equations, as follows [24,25,26]:(5)$$\nabla \times \vec{H}=\vec{J_{0}}$$
(6)$$\nabla \cdot \vec{B}=0$$
(7)$$\vec{B}=\mu_{0}\left(\vec{H}+\vec{M}\right)$$
(8)$$\vec{M}=\chi_{n}\vec{H}+\vec{M_{r}}$$
where $\vec{H}$ is the magnetic field intensity, $\vec{J_{0}}$ is the current density, $\vec{B}$ is the magnetic flux density, $\vec{M}$ is the magnetization, $\chi_{n}$ is the magnetic susceptibility, and $\vec{M_{r}}$ is the remnant magnetization of the permanent magnet.

#### 2.2.2. Acoustic Analysis Model of the BLDC Motor

After electromagnetic analysis, noise simulation was performed by mapping the electromagnetic force in the stator tooth, as illustrated in Figure 5. The number of elements used in the analysis was 88,465, and the number of nodes was 27,750.

Figure 6 presents the overall model of the analysis. The air medium encompassed the BLDC motor, and the measurement surface was set 100 mm away from the motor. 

The governing equation for the acoustic field is the following Helmholtz equation [27]:(9)$$\nabla^{2}P+\frac{k_{a}^{2}}{\rho_{0}}P=0$$
where P is the acoustic pressure, $k_{a}\;\left(\omega /c_{0}\right)$ is the acoustic wave number, $\rho_{0}$ (1.2 kg/m^3^) is the air density, and $c_{0}$ (343 m/s) is the acoustic velocity of air.

Somerfield’s boundary condition was used as the boundary condition of the medium so that no reflection would occur at the medium boundary, whereas the energy propagated out of the boundary. The boundary condition is expressed as follows [28]:(10)$$\nabla P\cdot n+ik_{a}P=0$$
where n is the unit normal vector at the boundary.

## 3. Results 

### 3.1. Numerical Analysis and Verification

Acoustic simulation and measurements were performed to analyze the noise generated by the geometry of the BLDC motor. To verify the simulation model, a prototype BLDC motor was fabricated, and the motor noise was measured in an anechoic environment as shown in Figure 7a. The BLDC motor noise measurement equipment consisted of a B&K 4165 microphone, B&K microphone multiplexer microphone amplifier, and LMS analysis equipment, as displayed in Figure 7b. To compare the experiment to the analysis under the same conditions, the noise was measured at an applied voltage of 13.5 V and a rotational speed of 1000 rpm. The measurement and simulation results in the noise of BLDC motor are shown in Figure 8. 

As shown in Figure 8a, the maximum value (23.48 dB) occurred at 3.15 kHz of the one-third octave band in the simulation results. At low frequencies, the sound pressure level (SPL) gradually increased with the increase in the frequency before decreasing again at 3.15–4 kHz. Considering that the SPL in the 1.6 kHz to 5 kHz bands is high, it suggests that the noise of the BLDC motor is a sensitive noise in the audible frequency range. Hence, the noise of the motor is an important factor determining the performance of the motor, and research for optimization of the noise is necessary.

As illustrated in Figure 8b, the sound pressure measurement result from the experimental model revealed a maximum SPL of approximately 26.2 dB at 3.15 kHz, which was 3.0 dB higher than the result from the simulation model displayed in Figure 8a. Because the difference in the overall SPL over the measurement frequency band (0.8–5 kHz) between the analysis model and experimental model was within 3 dB, the results were consistent, and the reliability of the analysis model was ensured. Figure 9 shows the difference between the simulation model and the experimental model in the 0.8–5 kHz frequency band range. The differential of results from experiment and simulation was indicated to relatively high below 1 kHz. However, the region where the maximum size of noise is generated is the 3 kHz band, and maximum error was generated to 10%. In addition, since the trend of the noise level in each frequency domain was very similar, it shows that the simulation model can similarly reflect the experimental results. 

Figure 10 shows the acoustic mode and frequency of the sound field generated inside the rotor. The cause of the noise generated from the motor is predicted to be that the electromagnetic forces become an excitation source to excite the structure, and the noise is generated due to the sound field resonance inside the rotor. Since there are few methods that can experimentally measure electromagnetic forces, a simulation study is very important. The electromagnetic force is an important cause of noise but also affects torque performance. Therefore, a study was conducted on a design parameter capable of reducing noise while maintaining the performance of the existing motor torque.

### 3.2. Design of Experiment for Optimization

#### 3.2.1. Design Factor Selection and Fractional Design 

Several design parameters exist for noise reduction of BLDC motors, but spatial limitations based on the volume-to-performance satisfaction maintain torque performance given the same volume and minimize structural vibration and acoustic resonance caused by the electromagnetic excitation force. The optimization of noise has been studied in many studies, mainly using design of experiments [29,30]. Therefore, five design parameters were used to improve noise reduction based on the verified analysis model, as displayed in Figure 11: slot depth (SD), stator tooth width (STW), slot opening (SO), radial depth (RD), and undercut angle (UA). To analyze whether the five design parameters are significant for torque performance and noise reduction, a partial factorial design was used to preferentially select the significant parameters and ensure the efficiency of the analysis. For the partial factorial design of the five design parameters, 16 factorial designs were defined, as listed in Table 3, and the torque and SPL were set as response values for the analysis results.

The significance of the design parameter selection can be verified based on Table 4 and Table 5 by analyzing the influence of the main effects on the response values of torque and SPL. The F-value is the division of the Adj MS of each parameter by the Adj MS of the error, and the larger the value, the greater the effect of the parameter on the response value. In addition, the *p*-value decreases as the F-value increases, and the design parameter is considered to be significant when the significance level is 0.05. Therefore, as shown in the significance analysis results of the main effects in Table 4 and Table 5 and Figure 12, the slot depth and stator tooth width with *p* ≤ 0.05 were identified as significant parameters.

#### 3.2.2. Design Factor Response Surface Analysis and Optimization 

To optimize the design parameters with significance, a central composite design was used to model the curvature of the design variables. The central composite design is a general design method used for response surface design to efficiently estimate quadratic terms. Table 6 displays the response design of the design parameters determined by the effectiveness analysis of the design parameters, and the quadratic regression model response surface for the response design can be estimated using the following equation:(11)$$y=\beta_{0}+\sum_{i=1}^{n_{d}}\beta_{i}x_{i}+\sum_{i=1}^{n_{d}}\sum_{j\geq i}^{n_{d}}\beta_{ij}x_{i}x_{j}$$
where $x_{i}$ is the independent variable (design parameter), y is the dependent variable (result value), and $\beta_{i}$ and $\beta_{ij}$ are coefficients determined using the least squares method. For the complete factorial design of the two design parameters that were proven to be effective through the partial factorial design, 13 factorial designs were defined as presented in Table 6, and the torque and SPL were set as response values for the analysis results.

As presented in Table 7, the analysis results of the response values of the response design parameters revealed that the main effects of the linear and quadratic terms of the slot depth and the linear term of stator tooth width were significant for the SPL. The prediction model for the SPL is expressed as presented in Equation (12) below, and the R-sq and R-sq (adj) values of 99.78% and 99.62%, respectively, indicate that the prediction equation is highly reliable.
OASPL = 21.49 − 0.410 ∗ SD + 0.274 ∗ STW + 0.5122 ∗ SD^2^(12)

As listed in Table 8, in the case of torque, the linear and quadratic terms of the slot depth were significant parameters, and the following regression equation was derived: the R-sq and R-sq (adj) values of 91.71% and 85.78%, respectively, indicate that the prediction equation is highly reliable. Considering the margin for the existing torque of 0.215 N∙m, the range of the slot depth at 0.216 N∙m or higher is 2.39–4.23 mm.
Torque = 0.20941 + 0.00428 ∗ SD − 0.000781 ∗ SD^2^(13)

Therefore, response optimization was performed by considering the limiting condition that would maintain the torque performance at 0.216 N∙m and the objective function that would minimize the OASPL. Figure 13 displays an optimization desirability of 98%, indicating high reliability. In the sound pressure prediction model, the SPL decreased with the decrease in the slot depth value, and in the optimal model for noise reduction, the torque was maintained with the highest noise reduction effect when the slot depth and stator tooth width were 2.39 and 9 mm, respectively.

Therefore, as presented in Table 9, applying the optimal design variable values to the analysis model revealed that the SPL was approximately 23.2 dB when the motor torque performance was maintained, which indicated a reduction of 4.9 dB in the overall SPL compared to the standard design model. Additionally, in the simulation results using the optimal parameter level, the torque was maintained, whereas the SPL was lower than that of the standard model. In Figure 14, the optimal model revealed an overall low SPL at all frequencies. Similar to the standard model, the SPL graph of the optimal model revealed that the SPL gradually increased proportionately with the frequency at low frequencies before decreasing again at 3.15–4 kHz. This phenomenon was attributed to the resonance phenomenon in the sound field inside the rotor, as the noise was caused by the resonance of the sound field inside the motor, as displayed in Figure 15. In the sound pressure analysis results, no changes occurred in the sound field resonance frequency of the optimal model and the standard design model, and the stator thickness and stator mass increased with the decrease in the stator depth, which resulted in noise reduction. This phenomenon can be explained by the mass law in (14). Equation (14) indicates that the transmission loss increases by 6 dB when the mass doubles, and the SPL decreases as the transmission loss increases because a reciprocal relationship exists between them:(14)$$TL=20\log_{10}\left(\frac{\omega m}{2Z_{0}}\right)$$
where TL is the transmission loss, ω is the resonance frequency, m is the mass, and $Z_{0}$ is the medium impedance.

### 3.3. Probabilistic Method 

To analyze and design mechanical systems, probabilistic design methods are considered depending on the uncertainties in the design variables or system parameters. Typically, uncertainties of a system are described as changes during usage, tolerances in manufacturing, and errors during the assembly process. As such, tolerances can cause uncertainties in product functions and unexpected performance, and optimization through analysis using probabilistic methods is necessary to minimize product uncertainties from a reliability perspective.

#### 3.3.1. Tolerance Design for Current-Level Performance Prediction 

Process capacity indicates the degree to which the product quality varies during the production process, when the process is in a controlled state. Process capacity is expressed as design capacity (standard tolerance) divided by process capacity (sigma level), and the standard deviation in the variation of design variables is calculated by Equation (15) while assuming a process capacity of $C_{p}=1$ at an adequate process level of 3σ. The distribution of the uncertain errors of the design variables was defined by generating 1,000,000 random numbers based on the mean and standard deviation as follows:(15)$$C_{p}=\frac{USL-LSL}{6\sigma }$$
where $C_{p}$ is the process capacity, USL is the upper specification limit of tolerance, LSL is the lower specification limit of tolerance, and σ is the standard deviation. The smaller the variation in quality is, the higher the processing capacity is. By contrast, the greater the variation in quality is, the lower the process capacity is. This relationship can be expressed in the process capacity index, as presented in Table 10 [31].

As illustrated in Figure 16, the tolerances for slot depth and stator tooth width were 2.39 ± 0.2 and 9.00 ± 0.2, respectively, which were expressed in triangular distributions, because of difficulties in determining the process capacity level during the development stage.

The design parameters, as defined by the triangular probability distributions and regression model in Equation (12), were analyzed by using Monte Carlo simulation. The sensitivity level was indicated an overwhelmingly greater influence of slot depth than stator tooth width in both torque and OASPL.

Therefore, as listed in the performance distribution result for the uncertain process capacity level in Figure 17, the SPL was determined by controlling the deviation in the slot depth, and the confidence level that would ensure a sound pressure range of 23.00–23.56 dB was 98.96%

#### 3.3.2. Tolerance Design Optimization Performance Prediction 

The optimization of design tolerance could maximize the noise reduction of the BLDC motor through quality management during production and that motor noise could be reduced by optimizing the design tolerance of the slot depth from the result of uncertain process capacity. Therefore, from the utility perspective of manufacturing management of design tolerance, a variation range of 3σ can generally be considered for adequate process capacity.

Table 11 lists the mean and standard deviation at the 3σ design tolerance level for the slot depth, and a normal probability distribution can be assumed, as presented in Figure 18, by generating 1,000,000 random numbers.

The design tolerance optimization results were analyzed by assuming the slot depth as a normal probability distribution at the 3σ level and applying it to the noise prediction model in Equation (12). The SPL revealed a performance distribution of 23.00–23.50 dB, and the confidence level to ensure a SPL of 23.5 dB or less was approximately 99.76%, as shown in Figure 19. Therefore, an SPL of 23.5 dB or less could be sufficiently secured when the slot depth was tolerance controlled at ±0.05.

## 4. Conclusions

In this study, a design process for developing a high-torque and low-noise BLDC motor was developed and verified. The effectiveness and noise characteristics of the motor were analyzed through design parameter analysis, and the low-noise structure of the motor was optimized. Design parameters were selected through a partial factorial design and optimized through response surface design. In addition, the optimal design of the design parameter tolerances was performed using the statistical method and Monte Carlo analysis. It is expected that the proposed process will contribute to robust design in the design stage for noise issues, and the conclusions are summarized as follows:

The cause of the motor noise at 3 kHz was found to be resonance inside and outside the rotor. After performing the optimization, it shows that the acoustic modes generated around 3 kHz are relatively reduced.

Among the design parameters influencing the BLDC motor noise, the stator slot depth and stator tooth width were identified as effective parameters through the DoE method, and the noise decreased with a decrease in the stator slot depth or an increase in the stator tooth width.Because the sensitivity of the noise reduction effect indicated that the effect of the stator slot depth was dominant, control noise could be adjusted by varying the slot depth.In the optimization of the design parameters for motor noise reduction, the objective function to minimize the SPL, and the limiting condition of the design parameters resulted in a stator slot depth and stator tooth width of 2.39 and 9.00 mm to ensure a SPL of 23.2 dB. Thus, a noise reduction of approximately 4.9 dB is expected compared with the standard model at 3 kHz.For the optimization of design tolerance using the statistical analysis method, the confidence level was 99.76% at the effective quality management level of 3σ, and the motor noise could be managed at 23.5 dB or lower by controlling the design tolerance of the slot depth at ±0.05 mm.

## Figures and Tables

**Figure 1 sensors-23-02483-f001:**
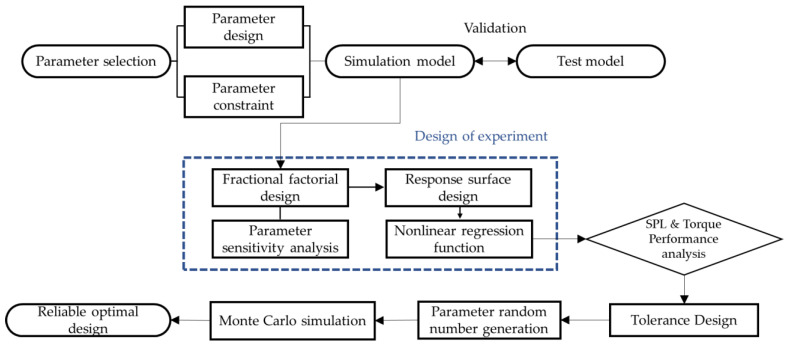
Flowchart of the design of the experiment.

**Figure 2 sensors-23-02483-f002:**
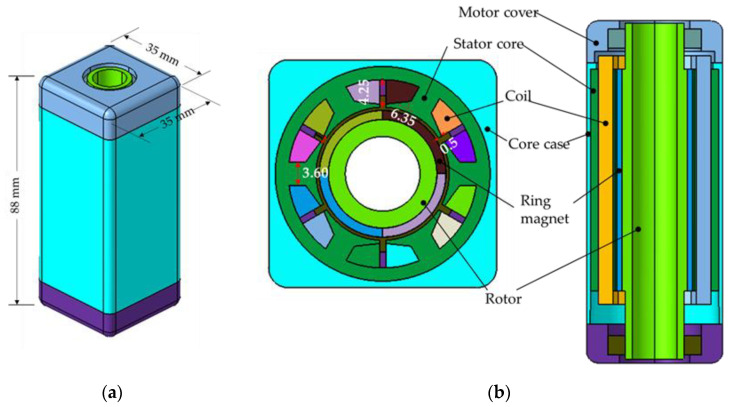
Design model of brushless direct-current (BLDC) motor: (**a**) isometric view model; (**b**) cross-section view model.

**Figure 3 sensors-23-02483-f003:**
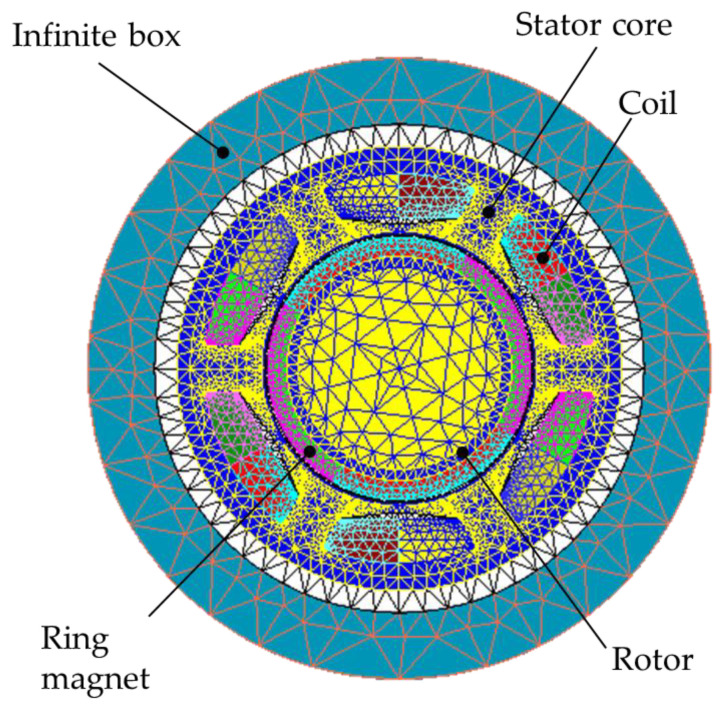
BLDC 2D model for electromagnetic simulation.

**Figure 4 sensors-23-02483-f004:**
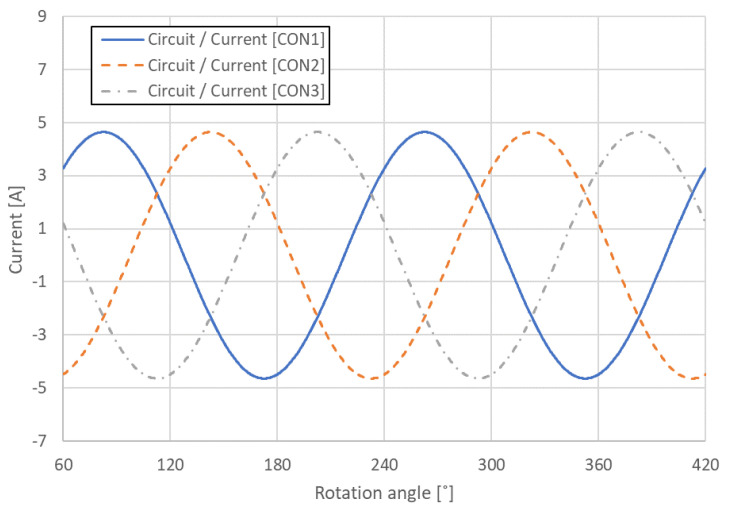
Current condition.

**Figure 5 sensors-23-02483-f005:**
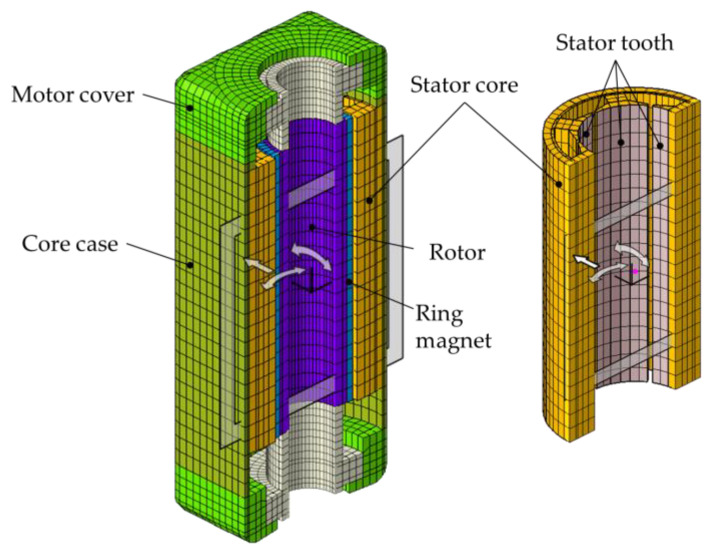
BLDC 3D model half-cut view for noise simulation.

**Figure 6 sensors-23-02483-f006:**
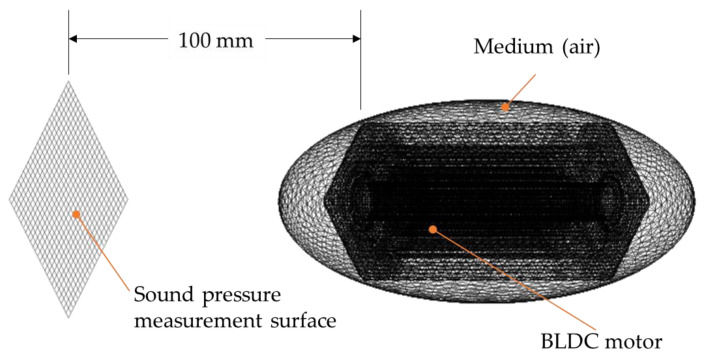
BLDC noise simulation setup.

**Figure 7 sensors-23-02483-f007:**
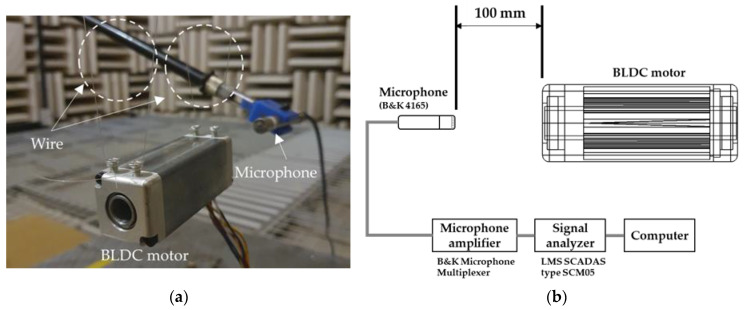
Test setup: (**a**) BLDC motor with the wire boundary condition; (**b**) schematic.

**Figure 8 sensors-23-02483-f008:**
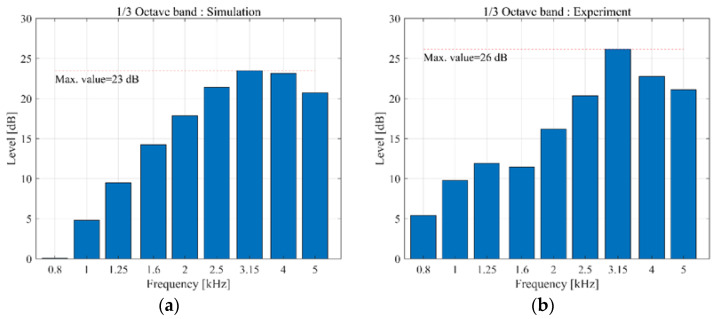
Sound pressure level for the standard model: (**a**) simulation results and (**b**) experimental results.

**Figure 9 sensors-23-02483-f009:**
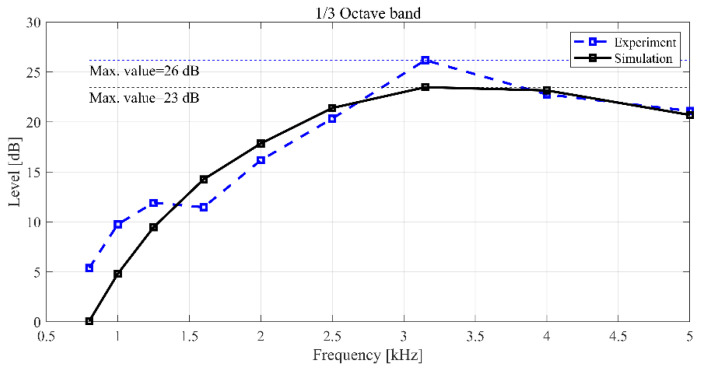
Comparison of experimental and simulation model of sound pressure level.

**Figure 10 sensors-23-02483-f010:**
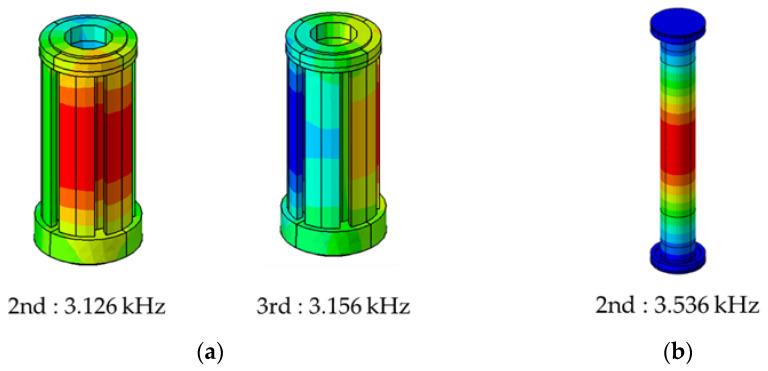
Acoustic mode for the standard model: (**a**) outside rotor and (**b**) inside rotor.

**Figure 11 sensors-23-02483-f011:**
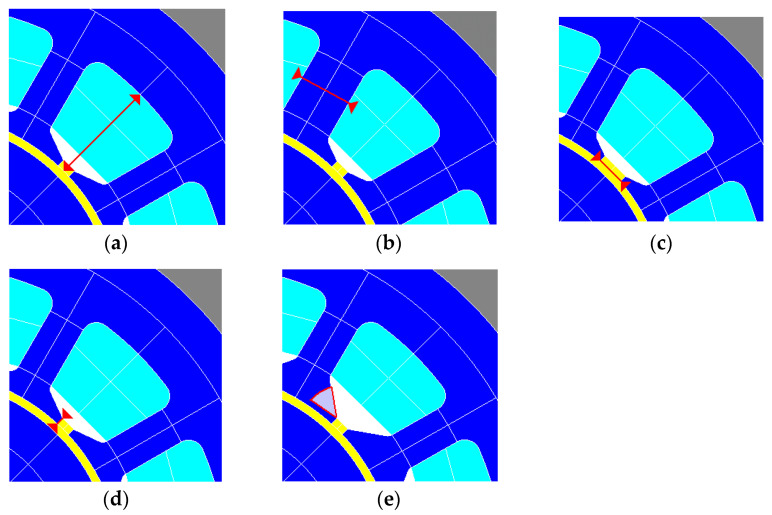
Selection factors for stator core design optimization: (**a**) slot depth; (**b**) stator tooth width; (**c**) slot opening; (**d**) radial depth; (**e**) undercut angle.

**Figure 12 sensors-23-02483-f012:**
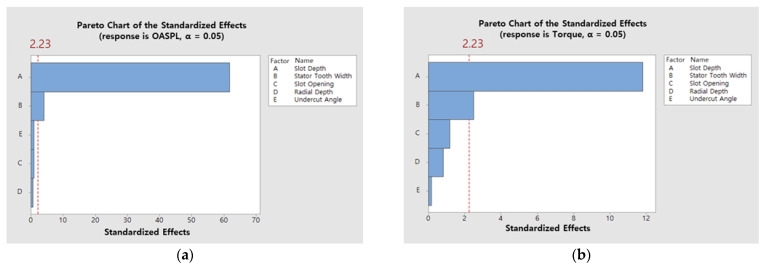
Result of fractional factorial design: (**a**) Pareto chart of the standardized effects of the OASPL; (**b**) Pareto chart of the standardized effects of the torque.

**Figure 13 sensors-23-02483-f013:**
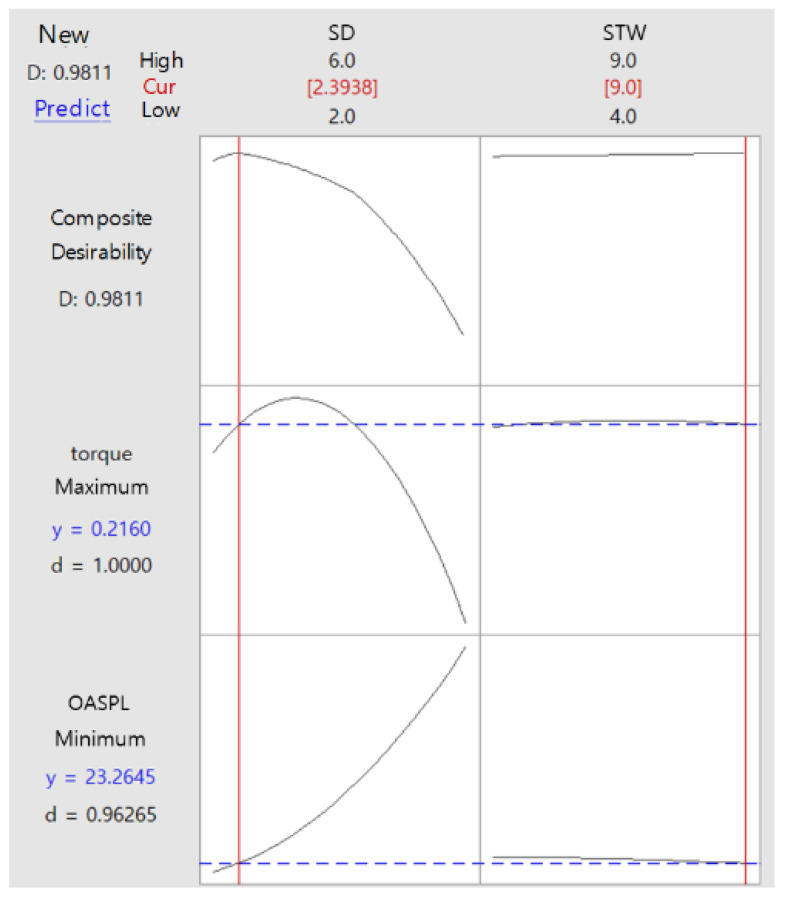
Desirability chart of design variable for optimization to maximize torque and minimize OASPL.

**Figure 14 sensors-23-02483-f014:**
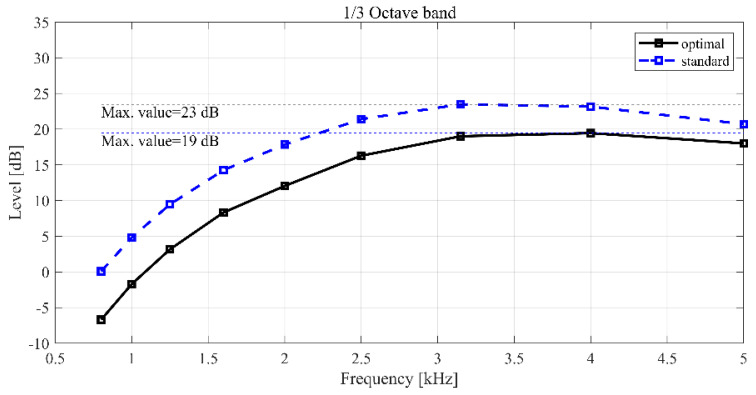
Comparison of the standard and optimal models of SPL.

**Figure 15 sensors-23-02483-f015:**
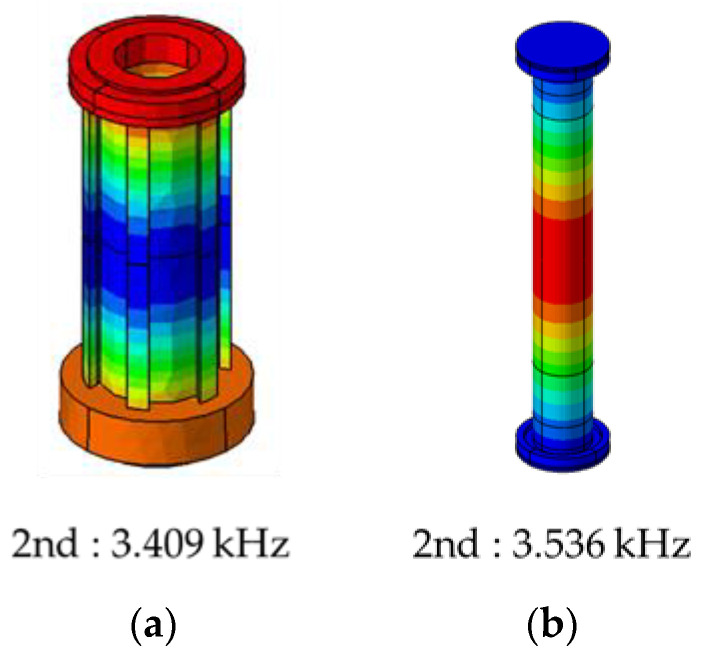
Acoustic mode for the optimal model: (**a**) outside rotor and (**b**) inside rotor.

**Figure 16 sensors-23-02483-f016:**
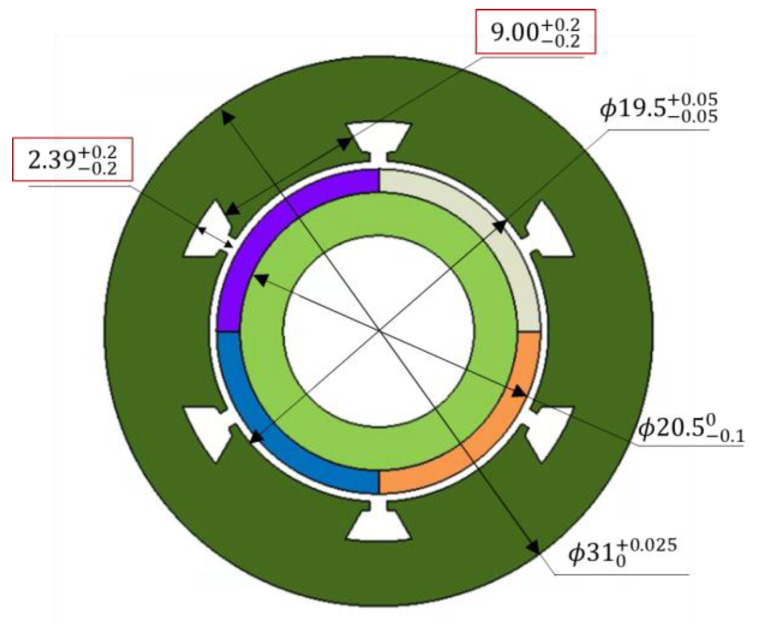
Design tolerance of the design parameters.

**Figure 17 sensors-23-02483-f017:**
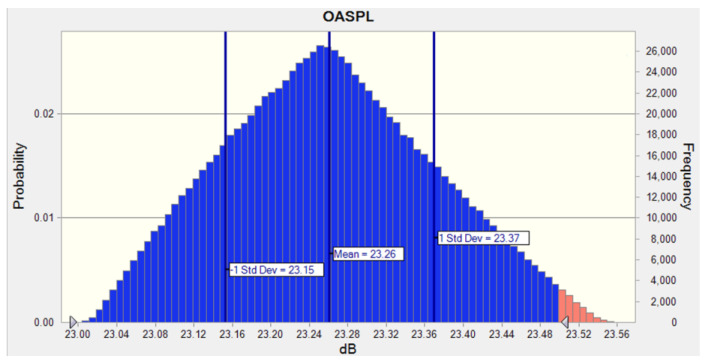
Performance distribution of probability triangular distribution for OASPL.

**Figure 18 sensors-23-02483-f018:**
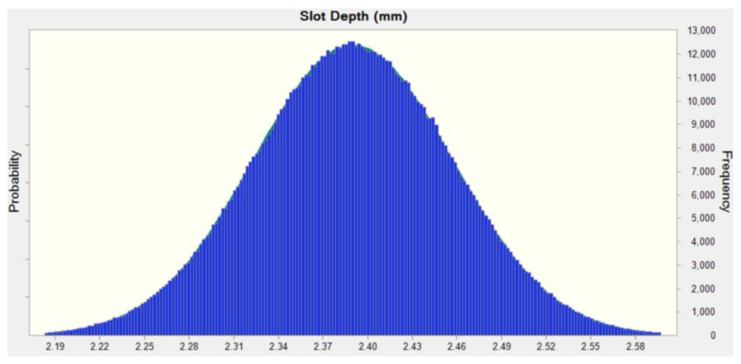
Probability normal distribution for the slot depth.

**Figure 19 sensors-23-02483-f019:**
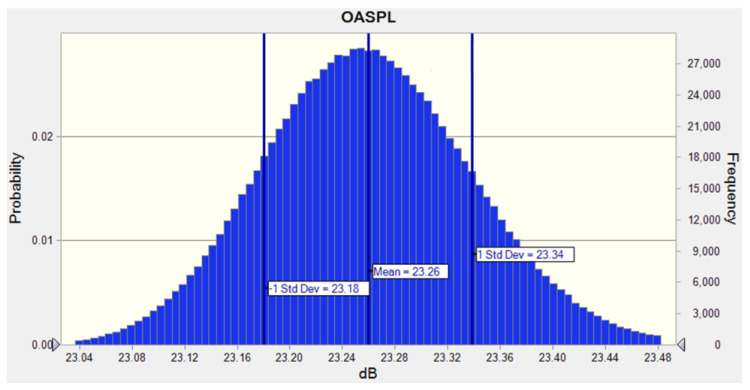
Performance distribution of probability normal distribution for OASPL.

**Table 1 sensors-23-02483-t001:** BLDC motor specifications.

Parameter	Value
Input voltage	13.5 V
Connection	3-phase
Number of poles/slots	4/6
Resistance of phase coil	0.209 Ω
Number of turns	15
Diameter of stator core	33 mm
Diameter of rotor core	16.7 mm

**Table 2 sensors-23-02483-t002:** Properties of the parts of the BLDC motor.

	Material	Density [kg/m^3^]	Young’s Modulus [GPa]	Poisson’s Ratio	Relative Permeability	Remanent Flux Density [T]
Core case	ZAMAK 2	6600	87	0.27		
Motor cover	AL6061	2700	68.9	0.33		
Stator	50PN470	7700	201	0.3	3980	
Rotor	S20C	7870	186	0.29	1085	
Ring magnet	NdFeB	7600	150	0.24	1.05	1.31

**Table 3 sensors-23-02483-t003:** Factors of fractional factorial design and results.

	Factor					Result	
	Slot Depth [mm]	Stator Tooth Width [mm]	Slot Opening [mm]	Radial Depth [mm]	Undercut Angle [°]	OASPL [dB]	Torque [N∙m]
Standard model	4.25	3.60	1	0.5	6.35	28.18	0.215
Case1	6	5	0.5	0.75	3	35.23	0.205
Case2	4	2	1.5	0.25	3	25.96	0.212
Case3	6	5	1.5	0.25	3	35.18	0.207
Case4	6	2	0.5	0.75	9	35.73	0.202
Case5	4	5	0.5	0.75	9	25.82	0.216
Case6	6	5	0.5	0.25	9	35.14	0.208
Case7	6	2	1.5	0.25	9	35.73	0.205
Case8	4	5	0.5	0.25	3	25.70	0.216
Case9	6	2	0.5	0.25	3	35.69	0.205
Case10	4	2	0.5	0.75	3	26.06	0.215
Case11	4	5	1.5	0.75	3	25.70	0.214
Case12	4	2	1.5	0.75	9	26.09	0.213
Case13	6	5	1.5	0.75	9	35.19	0.207
Case14	4	5	1.5	0.25	9	25.75	0.213
Case15	6	2	1.5	0.75	3	35.77	0.204
Case16	4	2	0.5	0.25	9	26.13	0.215

**Table 4 sensors-23-02483-t004:** Dispersion analysis of fractional factorial design for OASPL.

Source	DF	Adj SS	Adj MS	F-Value	*p*-Value
Model	5	337.803	67.561	766.52	0.000
Linear	5	337.803	67.561	766.52	0.000
SD	1	336.172	336.172	3814.07	0.000
STW	1	1.440	1.440	16.34	0.002
SO	1	0.065	0.065	0.74	0.410
RD	1	0.042	0.042	0.48	0.506
UA	1	0.084	0.084	0.95	0.352
Error	10	0.881	0.088		
Total	15	338.685			

**Table 5 sensors-23-02483-t005:** Dispersion analysis of fractional factorial design for torque.

Source	DF	Adj SS	Adj MS	F-Value	*p*-Value
Model	5	0.000334	0.000067	29.51	0.000
Linear	5	0.000334	0.000067	29.51	0.000
SD	1	0.000315	0.000315	139.25	0.000
STW	1	0.000014	0.000014	6.22	0.032
SO	1	0.000003	0.000003	1.35	0.272
RD	1	0.000002	0.000002	0.69	0.425
UA	1	0.000000	0.000000	0.03	0.871
Error	10	0.000023	0.000002		
Total	15	0.000356			

**Table 6 sensors-23-02483-t006:** Factors of central composite design and results.

	Factor		Result	
	Slot Depth [mm]	Stator Tooth Width [mm]	OASPL [dB]	Torque [N∙m]
Case1	4.00	6.50	27.21	0.216
Case2	6.00	6.50	36.01	0.209
Case3	4.00	9.00	26.81	0.216
Case4	5.41	8.27	31.90	0.214
Case5	4.00	6.50	27.28	0.216
Case6	4.00	6.50	27.28	0.216
Case7	2.59	4.73	24.06	0.216
Case8	2.59	8.27	23.74	0.216
Case9	4.00	4.00	27.73	0.215
Case10	5.41	4.73	33.02	0.213
Case11	4.00	6.50	27.28	0.216
Case12	2.00	6.50	22.77	0.216
Case13	4.00	6.50	27.28	0.216

**Table 7 sensors-23-02483-t007:** Dispersion analysis of central composite design for OASPL.

Source	DF	Adj SS	Adj MS	F-Value	*p*-Value
Model	5	169.204	33.841	635.07	0.000
Linear	2	161.540	80.770	1515.75	0.000
SD	1	160.601	160.601	3013.88	0.000
STW	1	0.939	0.939	17.63	0.004
Square	2	7.504	3.752	70.41	0.000
SD ∗ SD	1	7.302	7.302	137.02	0.000
STW ∗ STW	1	0.009	0.009	0.16	0.697
Interaction	1	0.160	0.160	3.00	0.127
SD ∗ STW	1	0.160	0.160	3.00	0.127
Error	7	0.373	0.053		
Total	12	169.577			

**Table 8 sensors-23-02483-t008:** Dispersion analysis of central composite design for torque.

Source	DF	Adj SS	Adj MS	F-Value	*p*-Value
Model	5	0.000046	0.000009	15.48	0.001
Linear	2	0.000028	0.000014	24.03	0.001
SD	1	0.000028	0.000028	46.84	0.000
STW	1	0.000001	0.000001	1.23	0.304
Square	2	0.000017	0.000009	14.45	0.003
SD ∗ SD	1	0.000017	0.000017	28.67	0.001
STW ∗ STW	1	0.000000	0.000000	0.05	0.837
Interaction	1	0.000000	0.000000	0.42	0.537
SD ∗ STW	1	0.000000	0.000000	0.42	0.537
Error	7	0.000004	0.000001		
Total	12	0.000050			

**Table 9 sensors-23-02483-t009:** Comparison of standard model and optimal model.

	Factor		Result	
	Slot Depth [mm]	Stator Tooth Width [mm]	OASPL [dB]	Torque [N∙m]
Standard model	4.25	3.60	28.18	0.215
Optimal model (DoE)	2.39	9.00	23.26	0.216
Optimal model (Simulation)	2.39	9.00	23.22	0.215

**Table 10 sensors-23-02483-t010:** Process capacity evaluation index.

	Estimation of the Process
$C_{p}$ < 1	Not adequate
$1.00\;\leq \;C_{p}$ < 1.33	Adequate
$C_{p}$ ≥ 1.33	Satisfactory enough
$C_{p}$ ≥ 1.66	Very satisfactory

**Table 11 sensors-23-02483-t011:** Mean and standard deviation for the design parameters.

Design Parameter	Mean	Standard Deviation
Slot Depth	2.39	0.067

## Data Availability

None.

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
