# Peer review of "A Study on Optimization of Noise Reduction of Powered Vehicle Seat Movement Using Brushless Direct-Current Motor"

_sensors, 2023, doi:10.3390/s23052483_

Round 1
Reviewer 1 Report
The article by Lee et.al. presents the fundamental concepts to be into consideration during the design of the seats in next-generation HEVs. The data and results are adequately presented, thus making understanding much better. As such, the article can be considered for publication after minor corrections based on the following points:
1. Affiliations are incomplete, therefore must be corrected.
2. References could be improved, given that most of the refences are based on the DoE methods, rather than what has been done in literature, especially on the noise reduction in HEVs. Refs such as Li et.al, Machines, 2022, 10, 670 and/or Yu et.al, Shock and Vibreation, 2017, 4132092, 1-13, could be added.
3. Font for reference section must be adjusted. Similarly to that of lines 173-181.
4. With so many figures in the articles, some of them could be put in the supplementary information, especially those that are less mentioned or discussed in text such as figures 9,17, or 18.
5.Line 165 should be figure 9 in stead of figure 10.
6. Line 161, consistency in recording of the frequencies is recommended.
7.Given the obtained results from both theoretical and experimental data on the SPL, what are the typical or literature SPL values within the audible frequency range?
8. From section 2.3, was the prototype fabricated based on the materials listed in table 1 or was it a commercially bought product?
9.Line 365, the recommended word could be 'adjusted' in place of 'controlled' so as to prevent confused meaning of the point #2.
Author Response
Reviewer #1
Thank you for your review.
The article by Lee et.al. presents the fundamental concepts to be into consideration during the design of the seats in next-generation HEVs. The data and results are adequately presented, thus making understanding much better. As such, the article can be considered for publication after minor corrections based on the following points:
Q1. Affiliations are incomplete, therefore must be corrected.
A1. Affiliations were modified.
“AI & Mechanical System Center, Institute for Advanced Engineering, Youngin-si 17180, Korea; leehj@iae.re.kr (H.-J. Lee); jaehyeon@iae.re.kr (J.-H. Nam)”
Q2. References could be improved, given that most of the refences are based on the DoE methods, rather than what has been done in literature, especially on the noise reduction in HEVs. Refs such as Li et.al, Machines, 2022, 10, 670 and/or Yu et.al, Shock and Vibreation, 2017, 4132092, 1-13, could be added.
A2. References were added.
“ 29. Li, T.; Wang, M.; He, Y.; Wang, N.; Yang, J.; Ding, R.; Zhao, K. Vehicle engine noise cancellation based on a multi-channel fractional-order active noise control algorithm. Machines, 2022,10,8, 670.
- Yu, H.; Zhang, X.; Zhang, C. Optimization Method of the Car Seat Rail Abnormal Noise Problem Based on the Finite Element Method. Shock Vib, 2017, 2017. “
Q3. Font for reference section must be adjusted. Similarly to that of lines 173-181.
A3. Font was modified in reference section and main text.
Q4. With so many figures in the articles, some of them could be put in the supplementary information, especially those that are less mentioned or discussed in text such as figures 9,17, or 18.
A4. Figures 17 and 18 were deleted. However, Figure 9 is important because it is a relative comparison result of experiment and FE analysis. Hence, the sentence was added as follows:
“The differential of results from experiment and simulation was indicated to relatively high below 1 kHz. However, the region where the maximum size of noise is generated is the 3 kHz band, and maximum error was generated to 10 %. In addition, since the trend of the noise level in each frequency domain was very similar, it shows that the simulation model can similarly reflect the experimental results.”
Q5. Line 165 should be figure 9 in stead of figure 10.
A5. Line 165 : the sentence was modified as follows :
“Figure 9 shows the difference ….”
Q6. Line 161, consistency in recording of the frequencies is recommended.
A6. Line 161: The unit of the frequencies was modified to kHz
“ … 26.2 dB at 3.15 kHz, …”
Q7. Given the obtained results from both theoretical and experimental data on the SPL, what are the typical or literature SPL values within the audible frequency range?
Q7. The noise of a motor is influenced by so many factors, such as applied voltage, operating speed, load conditions, magnetic flux density, and geometry. Also, the motor used in this paper developed as a motor for electric vehicles, it is difficult to refer to other literature specific data. However, it has been developed with very quiet performance compared to commercial motors.
Q8. From section 2.3, was the prototype fabricated based on the materials listed in table 1 or was it a commercially bought product?
A8. The motor used in this paper is a motor developed for seat of electric vehicle based on the materials listed in Table 1.
Q9. Line 365, the recommended word could be 'adjusted' in place of 'controlled' so as to prevent confused meaning of the point #2.
A9. The word was modified to recommended word.
“…control noise could be adjusted by varying the slot depth.”

Reviewer 2 Report
The authors proposed the article titled “A Study on Optimization of Noise Reduction of Powered Vehicle Seat Movement using Brushless Direct-Current Motor”. The following comments should be incorporated into the manuscript:
1. Figure 1 must be improved. All figures should be redrawn.
2. The Graphical representation of proposed model should be abstracted in the introduction section.
3. Organization of manuscript must be included in the last Para of introduction section.
4. The manuscript needs to include abbreviations.
5. I have not seen the novelty in proper form. The manuscript should emphasize novelty in a paragraph.
6. Latest relevant literature should be incorporated into the Manuscript.
7. The comparison through graph needs to be added to the results section.
8. Conclusions should be improved as per the results obtained.
Author Response
Reviewer #2
Thank you for your review.
The authors proposed the article titled “A Study on Optimization of Noise Reduction of Powered Vehicle Seat Movement using Brushless Direct-Current Motor”. The following comments should be incorporated into the manuscript:
Q1. Figure 1 must be improved. All figures should be redrawn.
A1. Figure1 was modified. Overall, all figures have been clearly modified.
Q2. The Graphical representation of proposed model should be abstracted in the introduction section.
A2. The abstract was modified as follows:
“In this paper, an optimal design model was developed to reduce noise and secure torque performance of BLDC motors used in seat of autonomous vehicle. An acoustic model using finite elements was developed and verified through the noise test of BLDC motor. In order to BLDC motor noise reduction and obtain a reliable optimization geometry of noiseless seat motion, parametric analysis was performed through design of experiment and Monte Carlo statistical analysis. The slot depth, stator tooth width, slot opening, radial depth, and undercut angle of the BLDC motor were selected as design parameters for design parameter analysis. Then, a non-linear prediction model was used to determine the optimal slot depth and stator tooth width to maintain the drive torque and minimize the sound pressure level at 23.26 dB or lower. The Monte Carlo statistical method was used to minimize the deviation of the sound pressure level caused by the production deviation of the design parameters. The results that the SPL was 23.00–23.50 dB with a confidence level of approximately 99.76% when the level of production quality control was set at 3σ.”
Q3. Organization of manuscript must be included in the last Para of introduction section.
A3. The organization of the thesis was modified while conducting Q7.
Q4. The manuscript needs to include abbreviations.
A4. The main script was revised overall.
Q5. I have not seen the novelty in proper form. The manuscript should emphasize novelty in a paragraph.
A5. The novel point of this thesis was written by revising the last paragraph of the introduction as follows:
“In this study, a design analysis process was proposed and verified to develop a seat motor for electric vehicles that improves motor drive torque and minimizes noise. Noise was measured through experiments, and the simulation model was verified based on frequency analysis. Partial factorial design was used to identify parameters with low influence, and response surface design was used for optimization. Noise optimization analysis was performed based on the acoustic mode. Additionally, Monte Carlo simulations were used to analyze the performance of the optimization for tolerance. Therefore, a process for noise optimization of BLDC motors was developed, and the proposed process is shown in Figure 1.”
Q6. Latest relevant literature should be incorporated into the Manuscript.
A6. Reference was added as follows:
“ 1. Karnavas, Y. L.; Topalidis, A. S.; Drakaki, M. Development and Implementation of a Low Cost μC-Based Brushless DC Motor Sensorless Controller: A Practical Analysis of Hardware and Software Aspects. Electronics, 2019, 8, 1456.
- Mohanraj, D.; Aruldavid, R.; Verma, R.; Sathyasekar, K.; Barnawi, A. B.; Chokkalingam, B.; Mihet-Popa, L. A review of BLDC Motor: State of Art, advanced control techniques, and applications. IEEE Access. 2022, 10, 54833-54869
- Sikora, A.; Zielonka, A.; Woźniak, M. Minimization of energy losses in the BLDC motor for improved control and power supply of the system under static load. Sensors. 2022, 22, 1058.
- Kumar, A.; Gandhi, R.; Wilson, R.; Roy, R. Analysis of permanent magnet BLDC motor design with different slot type. In 2020 IEEE International Conference on PESGRE, January 2020, pp. 1-6.
- Basu, S.; Rani, S. L. Generalized acoustic Helmholtz equation and its boundary conditions in a quasi 1-D duct with arbitrary mean properties and mean flow. J. Sound Vib. 2021, 512, 116377
- Li, T.; Wang, M.; He, Y.; Wang, N.; Yang, J.; Ding, R.; Zhao, K. Vehicle engine noise cancellation based on a multi-channel fractional-order active noise control algorithm. Machines, 2022,10,8, 670.
- Yu, H.; Zhang, X.; Zhang, C. Optimization Method of the Car Seat Rail Abnormal Noise Problem Based on the Finite Element Method. Shock Vib, 2017, 2017. “
Q7. The comparison through graph needs to be added to the results section.
A7. The structure of the thesis was modified as follows:
- Introduction
- Simulation model of BLDC Motor
2.1. BLDC Motor Design Model
2.2. Analysis Model of the BLDC Motor
2.2.1. Magnetic Analysis Model of the BLDC Motor
2.2.2. Acoustic Analysis Model of the BLDC Motor
- Results
3.1 Numerical analysis and Verification
3.2. Design of Experiment for Optimization
3.2.1. Design Factor Selection and Factorial Design
3.2.2. Design Factor Response Surface Analysis and Optimization
3.3. Probabilistic method
3.3.1. Tolerance Design for Current-Level Performance Prediction
3.3.2. Tolerance Design Optimization Performance Prediction
- Conclusions
Q8. Conclusions should be improved as per the results obtained.
A8. Conclusion was modified as follows:
“In this study, a design process for developing a high-torque and low-noise BLDC motor was developed and verified. The effectiveness and noise characteristics of the motor were analyzed through design parameter analysis, and the low-noise structure of the motor was optimized. Design parameters were selected through a partial factorial design and optimized through response surface design. In addition, the optimal design of the design parameter tolerances was performed using the statistical method and Monte Carlo analysis. It is expected that the proposed process will contribute to robust design in the design stage for noise issues, and the conclusions are summarized as follows:
The cause of the motor noise at 3 kHz was found to be resonance inside and outside the rotor. After performing the optimization, it shows that the acoustic modes generated around 3kHz are relatively reduced.
- Among the design parameters influencing the BLDC motor noise, the stator slot depth and stator tooth width were identified as effective parameters through the DoE method, and the noise decreased with a decrease in the stator slot depth or an increase in the stator tooth width.
- Because the sensitivity of the noise reduction effect indicated that the effect of the stator slot depth was dominant, control noise could be adjusted by varying the slot depth.
- In the optimization of the design parameters for motor noise reduction, the objective function to minimize the SPL and the limiting condition of the design parameters resulted in a stator slot depth and stator tooth width of 2.39 and 9.00 mm to ensure a SPL of 23.2 dB. Thus, a noise reduction of approximately 4.9 dB is expected compared with the standard model at 3 kHz.
- For the optimization of design tolerance using the statistical analysis method, the confidence level was 99.76% at the effective quality management level of 3σ, and the motor noise could be managed at 23.5 dB or lower by controlling the design tolerance of the slot depth at ±0.05 mm.”

Reviewer 3 Report
The paper is well prepared but needs to improve:
1.. The bibliographic references must be improved and updated, several recent papers from 2017 to 2023 must be cited. In his paper it is observed that more than half of the cited papers are very old.
2.. The formulation of the problem and the objectives of the research are not clear.
3.. A list of contributions of the paper should be made, please add it at the end of the introduction.
4.. Acronyms should be avoided in the abstract, please remove BLDC, SPL.
5.. Some equations do not have a bibliographic citation.
6.. The size of the letter and/or numbers should be increased in some figures, for example in figures (1, 5, 16, 17, 20, 21)
7.. The results graphs do not have a good presentation, background frames should be avoided, for example the figures (21, 20, 19, 17).
8.. Improve the presentation of the results of figures (12, 18).
9.. The conclusions should be improved, with respect to the objectives and contributions of the paper.
Author Response
Reviewer #3
Thank you for your review.
The paper is well prepared but needs to improve:
Q1. The bibliographic references must be improved and updated, several recent papers from 2017 to 2023 must be cited. In his paper it is observed that more than half of the cited papers are very old.
A1. Reference was added as follows:
“ 1. Karnavas, Y. L.; Topalidis, A. S.; Drakaki, M. Development and Implementation of a Low Cost μC-Based Brushless DC Motor Sensorless Controller: A Practical Analysis of Hardware and Software Aspects. Electronics, 2019, 8, 1456.
- Mohanraj, D.; Aruldavid, R.; Verma, R.; Sathyasekar, K.; Barnawi, A. B.; Chokkalingam, B.; Mihet-Popa, L. A review of BLDC Motor: State of Art, advanced control techniques, and applications. IEEE Access. 2022, 10, 54833-54869
- Sikora, A.; Zielonka, A.; Woźniak, M. Minimization of energy losses in the BLDC motor for improved control and power supply of the system under static load. Sensors. 2022, 22, 1058.
- Kumar, A.; Gandhi, R.; Wilson, R.; Roy, R. Analysis of permanent magnet BLDC motor design with different slot type. In 2020 IEEE International Conference on PESGRE, January 2020, pp. 1-6.
- Basu, S.; Rani, S. L. Generalized acoustic Helmholtz equation and its boundary conditions in a quasi 1-D duct with arbitrary mean properties and mean flow. J. Sound Vib. 2021, 512, 116377
- Li, T.; Wang, M.; He, Y.; Wang, N.; Yang, J.; Ding, R.; Zhao, K. Vehicle engine noise cancellation based on a multi-channel fractional-order active noise control algorithm. Machines, 2022,10,8, 670.
- Yu, H.; Zhang, X.; Zhang, C. Optimization Method of the Car Seat Rail Abnormal Noise Problem Based on the Finite Element Method. Shock Vib, 2017, 2017. “
Q2. The formulation of the problem and the objectives of the research are not clear.
A2. The last paragraph of the introduction was modified as follows:
“In this study, a design analysis process was proposed and verified to develop a seat motor for electric vehicles that improves motor drive torque and minimizes noise. Noise was measured through experiments, and the simulation model was verified based on frequency analysis. Partial factorial design was used to identify parameters with low in-fluence, and response surface design was used for optimization. Noise optimization analysis was performed based on the acoustic mode. Additionally, Monte Carlo simulations were used to analyze the performance of the optimization for tolerance. Therefore, a process for noise optimization of BLDC motors was developed, and the proposed process is shown in Figure 1.”
Q3. A list of contributions of the paper should be made, please add it at the end of the introduction.
A3. A list of the paper's contributions was compiled under conclusion.
“Author Contributions: Conceptualization, Hyun-Ju Lee and Jae-Hyeon Nam; methodology, Hyun-Ju Lee and Dong-Shin; software, Hyun-Ju Lee; validation, Hyun-Ju Lee and Dong-Shin Ko; formal analysis, Dong-Shin Ko; investigation, Hyun-Ju Lee; resources, Hyun-Ju Lee and Jae-Hyeon Nam; data curation, Hyun-Ju Lee; writing—original draft preparation, Hyun-Ju Lee and Dong-Shin Ko; writing—review and editing, Hyun-Ju Lee and Dong-Shin Ko; visualization, Hyun-Ju Lee; supervision, Dong-Shin Ko; project administration, Hyun-Ju Lee and Dong-Shin Ko; funding acquisition, Dong-Shin Ko. All authors have read and agreed to the published version of the manuscript.”
Q4. Acronyms should be avoided in the abstract, please remove BLDC, SPL.
A4. Acronyms was removed in the abstract.
“In this paper, an optimal design model was developed to reduce noise and secure torque performance of brushless direct-current motor used in seat of autonomous vehicle. An acoustic model using finite elements was developed and verified through the noise test of brushless direct-current motor. In order to brushless direct-current motor noise reduction and obtain a reliable optimization geometry of noiseless seat motion, parametric analysis was performed through design of experiment and Monte Carlo statistical analysis. The slot depth, stator tooth width, slot opening, radial depth, and undercut angle of the brushless direct-current motor were selected as design parameters for design parameter analysis. Then, a non-linear prediction model was used to determine the optimal slot depth and stator tooth width to maintain the drive torque and minimize the sound pressure level at 23.26 dB or lower. The Monte Carlo statistical method was used to minimize the deviation of the sound pressure level caused by the production deviation of the design parameters. The results that the SPL was 23.00–23.50 dB with a confidence level of ap-proximately 99.76% when the level of production quality control was set at 3σ.”
Q5. Some equations do not have a bibliographic citation.
A5. Reference was added as follows:
- Kumar, A.; Gandhi, R.; Wilson, R.; Roy, R. Analysis of permanent magnet BLDC motor design with different slot type. In 2020 IEEE International Conference on PESGRE, January 2020, pp. 1-6.
- Basu, S.; Rani, S. L. Generalized acoustic Helmholtz equation and its boundary conditions in a quasi 1-D duct with arbitrary mean properties and mean flow. J. Sound Vib. 2021, 512, 116377
Q6. The size of the letter and/or numbers should be increased in some figures, for example in figures (1, 5, 16, 17, 20, 21)
A6. The numbers, letter, and resolution of all figures in the paper was improved.
Q7. The results graphs do not have a good presentation, background frames should be avoided, for example the figures (21, 20, 19, 17).
A7. The background frames were deleted for the figures (21, 20, 19, 17).
Q8. Improve the presentation of the results of figures (12, 18).
A8. Figure 18 was deleted, and Figure 12 was enlarged.
Q9. The conclusions should be improved, with respect to the objectives and contributions of the paper.
A9. The conclusion was modified as follows:
“In this study, a design process for developing a high-torque and low-noise BLDC motor was developed and verified. The effectiveness and noise characteristics of the motor were analyzed through design parameter analysis, and the low-noise structure of the motor was optimized. Design parameters were selected through a partial factorial design and optimized through response surface design. In addition, the optimal design of the design parameter tolerances was performed using the statistical method and Monte Carlo analysis. It is expected that the proposed process will contribute to robust design in the design stage for noise issues, and the conclusions are summarized as follows:
The cause of the motor noise at 3 kHz was found to be resonance inside and outside the rotor. After performing the optimization, it shows that the acoustic modes generated around 3kHz are relatively reduced.
- Among the design parameters influencing the BLDC motor noise, the stator slot depth and stator tooth width were identified as effective parameters through the DoE method, and the noise decreased with a decrease in the stator slot depth or an increase in the stator tooth width.
- Because the sensitivity of the noise reduction effect indicated that the effect of the stator slot depth was dominant, control noise could be adjusted by varying the slot depth.
- In the optimization of the design parameters for motor noise reduction, the objective function to minimize the SPL and the limiting condition of the design parameters resulted in a stator slot depth and stator tooth width of 2.39 and 9.00 mm to ensure a SPL of 23.2 dB. Thus, a noise reduction of approximately 4.9 dB is expected compared with the standard model at 3 kHz.
- For the optimization of design tolerance using the statistical analysis method, the confidence level was 99.76% at the effective quality management level of 3σ, and the motor noise could be managed at 23.5 dB or lower by controlling the design tolerance of the slot depth at ±0.05 mm.”

Round 2
Reviewer 3 Report
Thank you very much for the first corrections:
1.. Author contributions are not required in the introduction, what is required is that you place in the introduction is a list of research contributions.
2.. The letters or words in figures 6, 7, 15 are very large and the numbers or letters in figures 12, 16 are very small or cannot be seen.
3.. If it is possible to improve the resolution of most of the figures since the letters and words lose sharpness when they are enlarged.
Author Response
Reviewer #3
Thank you for your review.
Thank you very much for the first corrections:
Q1. Author contributions are not required in the introduction, what is required is that you place in the introduction is a list of research contributions.
A1. The list of research contributions was added in the introduction as follows:
“The paper is structured as follows: in Section 2 BLDC Motor design model and simulation model are described; in Section3 numerical analysis for the motor noise is verified and Design of Experiment and probabilistic method is conducted for the optimization of the design; in Section 4, there is a discussion of results, followed by the conclusion.”
Q2. The letters or words in figures 6, 7, 15 are very large and the numbers or letters in figures 12, 16 are very small or cannot be seen.
A2. The figures were modified.
Q3. If it is possible to improve the resolution of most of the figures since the letters and words lose sharpness when they are enlarged.
A3. The figures used in the thesis were reviewed on the whole.
